# Silencing *KRIT1* Partially Reverses the Effects of Disturbed Flow on the Endothelial Cell Transcriptome

**DOI:** 10.3390/ijms26094340

**Published:** 2025-05-02

**Authors:** Amelia Meecham, Sara McCurdy, Eduardo Frias-Anaya, Wenqing Li, Helios Gallego-Gutierrez, Phu Nguyen, Yi-Shuan Li, Shu Chien, John Y.-J. Shyy, Mark H. Ginsberg, Miguel Alejandro Lopez-Ramirez

**Affiliations:** 1Department of Medicine, University of California, La Jolla, CA 92093, USA; ameecham@health.ucsd.edu (A.M.); srgmccurdy@gmail.com (S.M.); efriasanaya@health.ucsd.edu (E.F.-A.); liwenqing753@gmail.com (W.L.); hgallegogutierrez@health.ucsd.edu (H.G.-G.); 2Department of Bioengineering, University of California, La Jolla, CA 92093, USA; p8nguyen@ucsd.edu (P.N.); yili@ucsd.edu (Y.-S.L.); shuchien@ucsd.edu (S.C.); jshyy@health.ucsd.edu (J.Y.-J.S.); 3Department of Pharmacology, University of California, La Jolla, CA 92093, USA

**Keywords:** endothelium, atherosclerosis, pulsatile shear stress, oscillatory shear stress, vasoprotection, endothelial cells

## Abstract

Endothelial cells respond to forces generated by laminar blood flow with changes in vasodilation, anticoagulant, fibrinolytic, or anti-inflammatory functions which preserve vessel patency. These responses to flow shear stress are primarily mediated by the modulation of the following transcription factors: Krüppel-like factors 2 and 4 (KLF2 and KLF4). Notably, disturbed flow patterns, which are found in vascular areas predisposed to atherosclerosis, significantly reduce the endothelial expression of KLF2 and KLF4, resulting in changes in the transcriptome that exacerbate inflammation and thrombosis. The endothelial CCM (Cerebral Cavernous Malformation) complex, comprising KRIT1 (Krev1 interaction trapped gene 1), CCM2 (Malcavernin), and CCM3 (Programmed cell death protein 10), suppresses the expression of KLF2 and KLF4. Loss of function of the CCM complex has recently been suggested to protect from coronary atherosclerosis in humans. We thus hypothesized that the silencing of *KRIT1*, the central scaffold of the CCM complex, can normalize the atherogenic effects of disturbed flow on the human endothelial transcriptome. Bulk RNA sequencing (RNA-seq) was conducted on human umbilical vein endothelial cells (HUVECs) after the expression of KRIT1 was silenced using specific small interfering RNA (siRNA). The endothelial cells were exposed to three different conditions for 24 h, as follows: pulsatile shear stress (laminar flow), oscillatory shear stress (disturbed flow), and static conditions (no flow). We found that silencing the *KRIT1* expression in HUVECs restored the expression of the transcription factors KLF2 and KLF4 under oscillatory shear stress. This treatment resulted in a transcriptomic profile similar to that of endothelial cells under pulsatile shear stress. These findings suggest that inhibition of the CCM complex in endothelium plays a vasoprotective role by reactivating a protective gene program to help endothelial cells resist disturbed blood flow. Targeting CCM genes can activate well-known vasoprotective gene programs that enhance endothelial resilience to inflammation, hypoxia, and angiogenesis under disturbed flow conditions, providing a novel pathway for preventing atherothrombosis.

## 1. Introduction

Inflammation in cardiovascular diseases is a significant and common clinical issue, and morbidity is often due to thrombosis and endothelial dysfunction [1]. Healthy, quiescent vascular endothelial cells have anticoagulant, fibrinolytic, and anti-inflammatory properties [2,3]. Flow patterns are known to have specific effects on vascular endothelial gene expression, and have been the focus of numerous studies, highlighting the importance of specific genetic programs in maintaining endothelial homeostasis [4,5,6]. The vasoprotective effects of laminar blood flow are primarily driven by the upregulation of the transcription factors KLF2 and KLF4 (Krüppel-like factors 2 and 4) [7,8]. These factors significantly enhance the expression of genes encoding anticoagulants, such as *THBD*, which encodes thrombomodulin (TM), and vasodilators, like *NOS3*, which encodes endothelial nitric oxide synthase (eNOS). KLF2 and KLF4 suppress the expression of genes that inhibit angiogenesis, including *THBS1*, which encodes thrombospondin-1 (TSP1), as well as pro-inflammatory genes driven by NF-κB, such as vascular adhesion molecules *VCAM1* and *ICAM1* [7,9,10,11]. Moreover, KLF2 and KLF4 are regulators of hypoxia by promoting the degradation of hypoxia-inducible factor 1 (HIF-1) [12]. These vasoprotective effects of laminar blood flow have inspired new therapeutic strategies centered around enhancing the expression of endothelial KLF2 and KLF4 [7,13,14], to reduce cardiovascular morbidity [15,16].

Recent in vitro studies have documented the gene expression programs in cultured human umbilical vein endothelial cells (HUVECs) and human aortic endothelial cells (HAECs) exposed to different patterns of shear stress [17]. Pulsatile shear stress (laminar flow) promotes an atheroprotective endothelial phenotype (e.g., by increasing eNOS, Cholesterol-25-hydroxylase, Ch25h; Liver X receptor, LxR; cycle-dependent kinase 2, CDK2), while oscillatory shear stress (disturbed flow) is linked to an atheroprone phenotype (e.g., by increasing NF-ĸB signaling and oxidative stress) [6,17,18,19,20,21]. Using these culture systems, key mechanotransduction pathways vital for endothelial function and homeostasis have been identified [17,18,19,22,23,24,25]. For instance, oscillatory shear significantly downregulates KLF2 and KLF4 in endothelial cells, leading to the dysregulation of many downstream vasoprotective gene programs [19].

A recent study utilizing Perturb-seq, a method combining CRISPR-based gene silencing with single-cell RNA sequencing, investigated the effects of individually silencing 300 genes in aortic endothelial cells. These genes were either associated with or located near SNPs linked to an increased risk of coronary artery disease (CAD). The analysis showed that the Cerebral Cavernous Malformation (CCM) signaling pathway is connected to CAD risk genes and influences atheroprotective processes in endothelial cells [26].

CCMs are vascular lesions composed of clusters of abnormally leaky and fragile capillaries in the central nervous system, which can result in hemorrhagic stroke or seizures [27,28]. *KRIT1*, sometimes known as *CCM1*, is one of three genes whose loss drives the formation of these malformations [29]. Beyond its structural role, KRIT1 also regulates key signaling pathways, including those involved in inflammation and mechanosensitive responses to blood flow [30,31].

In this study, we tested the hypothesis that silencing *KRIT1* would promote vasoprotective transcriptional responses in endothelial cells exposed to disturbed flow. We performed bulk RNA sequencing (RNAseq) on HUVEC treated with *KRIT1* siRNA (siKRIT1) and exposed to pulsatile, oscillatory, or no flow for 24 h. Our findings indicate that targeting the endothelial CCM complex may help to restore the expression of vasoprotective gene programs downregulated by oscillatory shear stress. By mimicking a pulsatile shear stress gene program, this approach may help to negatively regulate inflammation, hypoxia, and angiogenesis in mature vascular endothelial cells, ultimately protecting against atherothrombosis.

## 2. Results

### 2.1. Transcriptomic Responses of KRIT1-Targeted HUVECs When Subjected to Oscillatory and Pulsatile Shear Stress or Static Conditions

To evaluate whether the silencing of *KRIT1* can enhance the expression of the transcription factors KLF2 and KLF4 in endothelial cells experiencing different patterns of shear stress, we subjected *KRIT1* siRNA-treated HUVECs to pulsatile shear (PS) flow, oscillatory shear (OS) flow, or no flow (static) for 24 h (Figure 1A). We observed that silencing *KRIT1* (siKRIT1) led to increased levels of the transcription factors *KLF2* and *KLF4*, as well as the KLF2/4 target gene, *NOS3,* across all conditions, as measured by RT-qPCR (Figure 1B). Conversely, downregulation of these important atheroprotective genes was observed in the siSCR control HUVECs subjected to oscillatory flow, compared to pulsatile shear conditions (Figure 1B). These data agree with previous studies reporting the same regulation of these flow-responsive genes under laminar (PS) or disturbed (OS) flow, both in vitro and in vivo [17]. Interestingly, targeting *KRIT1* expression (siKRIT1) in HUVECs exposed to oscillatory flow significantly restored expression of *KLF2* and *KLF4* and *NOS3* mRNA to levels observed in siSCR control cells under pulsatile shear flow (Figure 1B).

To better understand the transcriptomic profile of changes due to KRIT1-targeting, we performed bulk RNA sequencing (RNAseq) on the groups of HUVECs as described above; *KRIT1* knockdown (siKRIT1) vs. control (siSCR) subjected to pulsatile shear or oscillatory shear flow. Significant differentially expressed genes were determined by DESeq-adjusted *p* values (*p* < 0.05), and at least a log fold change of 1. We observed that KRIT1-targeted HUVECs under oscillatory shear flow exhibited a transcriptomic profile that closely resembled that of HUVECs treated with control siRNA under pulsatile shear flow (Figure 1C). Notably, consistent with RT-qPCR results, we observed that *KRIT1-*targeted HUVECs under oscillatory shear flow increased expression of transcription factors *KLF2* and *KLF4* (Figure 1D). Differential expression values of these genes can be found in Appendix A.

### 2.2. Silencing KRIT1 in Endothelial Cells Experiencing Disturbed Flow Activates Protective Gene Programs Known to Enhance Resistance to Inflammation, Hypoxia, and Angiogenesis

To investigate the transcriptional effects of KRIT1-targeted HUVECs, we conducted pathway analysis using the differentially expressed genes (DEGs) identified comparing siKRIT1 HUVECs to the siSCR HUVECs. A total of 59 pathways were found to be significantly enriched under oscillatory shear flow, including those related to cell-matrix adhesion, response to shear stress, integrin-mediated signaling, response to hypoxia, and regulation of angiogenesis (Figure 2A, Appendix A). For inflammation pathway genes, the expression profile of siKRIT1 cells exposed to oscillatory shear flow showed notable similarities to those of siSCR HUVEC under pulsatile shear flow, including the elevation of *CMKLR1*, *IL17RE*, *TBXA2R* and downregulation of *JUN*, *SPHK1*, *P2RX7* and *C2CD4B* (Figure 2B). In hypoxia pathways, the gene expression patterns in siKRIT1 HUVEC under oscillatory shear flow again aligned with those seen in siSCR HUVEC under pulsatile shear flow, including the elevation of *AQP1*, *ADRB2*, *HIF3A*, further supporting a potential protective transcriptional response against oscillatory shear flow-induced hypoxic stress. We also observed that with angiogenesis pathway genes, siKRIT1 HUVEC under oscillatory shear flow also showed expression changes similar to those seen in control cells under pulsatile shear flow. Notably, *CYP1B1*, *APLNR*, *RAMP2*, and *SEMA5A* were increased. These results indicate that *KRIT1* silencing in HUVEC exposed to disturbed flow leads to a change in transcriptome associated with vascular protection.

## 3. Discussion

We demonstrate that targeting endothelial KRIT1 initiates a protective response in endothelial cells exposed to disturbed flow patterns, which helps prevent the onset of an atheroprone gene program associated with endothelial dysfunction. In our study, we utilized flow chamber experiments to expose endothelial cells to various conditions, including oscillatory shear stress, pulsatile shear stress, and static conditions. Our analysis indicated that targeting *KRIT1* expression in HUVEC under these three conditions resulted in a significant increase in the levels of the transcription factors *KLF2* and *KLF4*, as well as in protective gene programs that enhance resilience to inflammation, hypoxia, and angiogenesis in the face of disturbed flow conditions.

Vascular endothelial cells undergo transcriptional changes in response to specific flow patterns. Exposure to oscillatory shear stress in vitro triggers transcriptomic responses mimicking those observed in the endothelium of athero-prone aortic regions. Likewise, endothelial cells subjected to pulsatile shear stress in vitro exhibit gene expression patterns resembling those found in vascular regions with laminar flow, which are known to be resistant to atherosclerotic plaque formation [32,33]. The flow-responsive transcription factors KLF2 and KLF4 play an important role in restricting endothelial dysfunction, inflammation, thrombosis, and angiogenesis in response to disturbed flow [34,35].

The link between the CCM complex and cardiovascular disease risk is the focus of a recent study by Schnitzler et al., responsible for identifying 43 coronary artery disease GWAS signals that all converged on the CCM complex pathway. The authors went on to show that CRISPR knockdown of *CCM2*, the main binding partner of KRIT1, in endothelial cells leads to the activation of anti-inflammatory, anti-thrombotic, and barrier-promoting genetic programs [26]. While KRIT1 is considered the central scaffold of the CCM protein complex, it also has a transmembrane binding partner, heart of glass 1 (HEG1), which is responsible for anchoring KRIT1 at endothelial junctions [15].

Our own recent work has shown that the interaction between KRIT1 and HEG1 can be pharmacologically inhibited, resulting in increased endothelial expression of *KLF2*, *KLF4*, and *eNOS* [15,36], establishing the endothelial CCM complex as a potentially druggable target, with small molecule inhibitors representing a potential therapeutic option for the prevention of endothelial dysfunction in vascular disease.

Pharmacologically inhibiting the endothelial KRIT1-HEG1 protein interaction with a compound denoted HKi2 [15] under static conditions increases the expression of *KLF4* and *KLF2* and mimics the vasoprotective transcriptional effects of laminar blood flow [15,36]. Here, we extend those studies in the following two ways: (1) we show that genetic silencing of *KRIT1* triggers similar transcriptional programs to those initiated by HKi2, and (2) we show that under conditions of disturbed flow, known to trigger transcriptional changes associated with predisposition to atherosclerosis, silencing *KRIT1* leads to a transcriptional program more like that observed in atheroprotective laminar flow.

These findings align with a recent study showing that blood flow can drive the release of endothelial KRIT1 from the CCM complex and trigger the activation of a MEKK3-MEK5-ERK5-MEF2 pathway [37] that increases KLF2 and KLF4 expression [37]. Furthermore, the HEG1 protein migrates to the downstream side of the cells and is secreted into the medium alongside KRIT1 [37]. Depleting KRIT1-HEG1 from endothelial cells triggers the activation of the MEKK3-MEK5-ERK5-MEF2 pathway, which is a well-known regulator of KLF2 and KLF4 expression [7,16,38]. Future research should focus on gaining a deeper understanding of the biology of the endothelial CCM protein complex. This understanding is essential to clarify the molecular mechanisms that differentiate between beneficial and protective processes and those that result in harmful dysfunction in endothelial cells associated with CCM disease [39,40].

Taken together, these studies, along with our findings, strengthen the emerging paradigm that KRIT1 regulates endothelial adaptation to disturbed flow through transcriptional regulation of vasoprotective pathways. Furthermore, they highlight the potential of CCM complex proteins as therapeutic targets for vascular disease.

## 4. Material and Methods

### 4.1. Cell Culture and Transfection with siRNA

HUVECs purchased from Lonza (C2519A, Walkersville, MD, USA) were cultured according to the manufacturer’s protocol and recommended growth medium (CC-3162). Experiments described were performed using cells between passages 3 and 6. HUVECs were maintained by passaging 1:3 and grown to confluence in fibronectin-coated T-75 flasks using 2 μg/mL human fibronectin (F2006 MilliporeSigma, St. Louis, MO, USA) dissolved in sterile PBS and incubated for 1 h at 37 °C. For in vitro experiments, cells were transfected with 75 nM SMARTpool siRNA against *KRIT1* (M-003825–01, Dharmacon/Horizon, Lafayette, Colorado, USA) or non-targeting scrambled control siRNA (D-001206–13, Dharmacon/Horizon, Lafayette, Colorado, USA). Transfection mix was prepared in OptiMEM (Gibco/Thermo Fisher Scientific, Waltham, MA, USA) using Lipofectamine3000 (Invitrogen/Thermo Fisher Scientific, Waltham, MA, USA), which was added dropwise to cells plated in complete growth medium. Following overnight (12-h) incubation, cells were given fresh growth medium for 24 h before replating onto fibronectin-coated glass slides for flow chamber experiments.

### 4.2. Flow Chamber Experiments

HUVECs transfected with (siRNA) for 48 h were replated using 0.05% Trypsin/EDTA (25300054, Thermo Fisher Scientific, Waltham, MA, USA) on 38 mm × 76 mm glass plates at 95% confluence. Previous studies in HUVECs have shown that siRNA treatment leads to a 70% reduction in KRIT1 protein expression after 48 h [41]. A subset of the glass plates was then assembled into a parallel-plate flow channel, as previously described [42] for each condition, as follows: siSCR+ pulsatile shear flow, siKRIT1+ pulsatile shear flow, siSCR+ oscillatory flow, and siKRIT1+ oscillatory flow. The flow system was maintained at 37 °C and ventilated with 95% humidified air with 5% CO_2_. HUVECs were exposed to pulsatile shear flow (PS) (12 ± 4 dyn/cm^2^, 1 Hz) or oscillatory flow (OS) (1 ± 4 dyn/cm^2^, 1 Hz) for 24 h. Additional control plates of siSCR- and siKRIT1-transfected HUVECs were subjected to static conditions (no flow) and maintained in the same culture medium and conditions described above in an adjacent incubator.

### 4.3. RNA Isolation

Glass slides of HUVECs were removed from the flow chamber and cells were directly lysed in 400 µL of TRIzol (Invitrogen, 15596026) per slide and stored in RNase-free microcentrifuge tubes, which were transferred to dry ice for storage until all samples were collected. Total RNA was extracted using the RNeasy mini kit (Qiagen, 74104) according to manufacturer’s directions. RNA quality and quantity were measured on a NanoDrop spectrophotometer (Thermo Fisher) before cDNA synthesis using the qScript cDNA SuperMix First-Strand cDNA Synthesis (Quantabio, 95048-100). Silencing of the *KRIT1* transcript and expression of known downstream genes was verified by quantitative real-time PCR (qRT-PCR) (primer sequences can be found in Appendix A) before samples were submitted to the UCSD IGM Genomics Center for RNA integrity analysis and bulk RNA sequencing.

### 4.4. RNA-Sequencing (RNA-Seq) Analysis

RNA quality was confirmed using an Agilent Tapestation 4200, and high-quality RNA (RIN > 8.0) was used for RNA-seq library preparation with the Illumina^®^ Stranded mRNA Prep kit (Illumina, San Diego, CA, USA). RNA libraries were multiplexed and sequenced with 100-bp paired reads (PE100) on an Illumina NovaSeq 6000 platform with a target depth of 30 million reads per sample. Samples were demultiplexed using bcl2fastq conversion software v2.20 (Illumina, San Diego, CA, USA).

Sequencing analysis was conducted in the R environment, following quality assurance of the data using FastQC v0.12.0. Reads were aligned to the human genome (GRCh38) using RNAStar v2.7.11a [43], and read counts were obtained using featureCounts v2.0.6 [44]. Differential expression analysis and normalization of counts were performed using DESeq2 v 1.32.0 [45]. RNA-seq data from this study are accessible in the Gene Expression Omnibus (GEO) database (GSE288811, http://www.ncbi.nlm.nih.gov/geo) (accessed on). The corresponding authors can be contacted for requests regarding public and non-public data supporting this study’s findings.

### 4.5. Statistical Analysis

Statistical analyses were performed using GraphPad Prism software, version 9.0. Data are presented as mean ± standard error of the mean (SEM) across multiple biological experiments, with the number of independent replicates specified for each experiment. For comparisons between groups, multiple unpaired t-tests using the Holm–Sidak method were used to adjust for multiple testing, with a significance threshold set at *p* < 0.05.

## Figures and Tables

**Figure 1 ijms-26-04340-f001:**
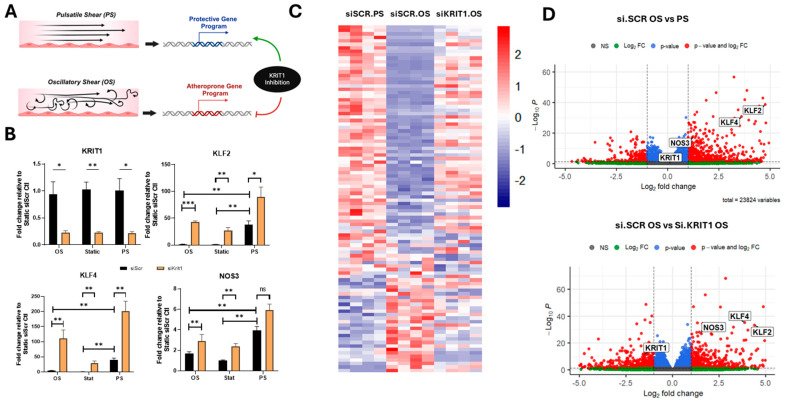
Transcriptomic responses of *KRIT1*-targeted HUVECs subjected to different flow patterns. (**A**) Schematic depicting the relationship between flow type and endothelial response, with KRIT1-targeting leading to a transcriptomic profile that is most similar to that seen in cells experiencing vasoprotective pulsatile flow. (**B**) RT-qPCR analysis of key vasoprotective genes (*KLF2*, *KLF4*, *NOS3*) and confirmation of *KRIT1* silencing in scrambled siRNA (siSCR)- or *KRIT1* siRNA (siKRIT1)-treated HUVECs exposed to no flow (static), pulsatile shear (PS), or oscillatory shear (OS) for 24 h. (*n* = 4, * *p* < 0.05, ** *p* < 0.001, *** *p* < 0.0001). (**C**) Heatmap of top 100 differentially expressed genes comparing PS to OS flow under control conditions following bulk RNAseq of the samples described in panel A (*n* = 4). (**D**) Volcano plots from bulk RNASeq. (**Top**) Comparison of PS and OS flow under control conditions highlighting higher expression of key vasoprotective genes (*KLF2*, *KLF4*, *NOS3*) under PS. (**Bottom**) Comparison of transcriptome in control and KRIT1 knockdown cells under OS, highlighting the successful knockdown of *KRIT1* and upregulation of *KLF2*, *KLF4*, and *NOS3* in siKRIT1-treated cells relative to siSCR control. Red points represent genes that changed > 2 fold and *p* < 0.05 green points indicate genes that changed > 2 fold but *p* > 0.05. Blue points indicate genes that changed < 2 fold. Gray points indicate non-significantly changed genes.

**Figure 2 ijms-26-04340-f002:**
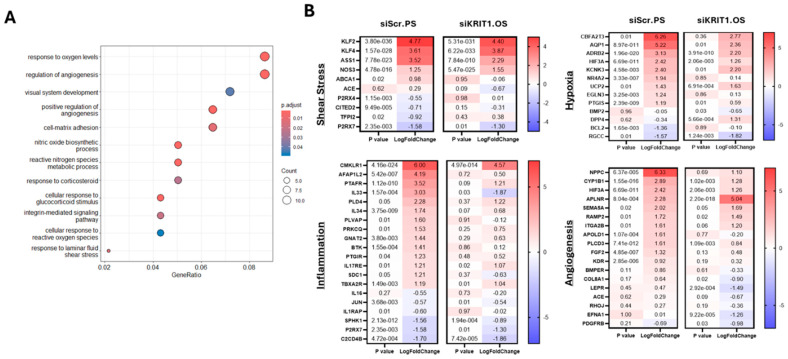
Biological pathway analysis of KRIT1-targeted HUVECs under disturbed flow conditions. (**A**) Dot plot highlighting a selection of the significantly enriched biological pathways based on the DEGs in response to OS (oscillatory shear) in both control and *KRIT1* knockout cells. Pathway analysis was performed using the ‘clusterProfiler’ package in R. (**B**) Heatmaps of selected genes from four of the significantly changed pathways. Log fold changes are comparable to siScr.OS condition. *p* value and LogFoldChange generated by Deseq2 package (*n* = 4).

## Data Availability

Data is contained within the article and Appendix A.

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
