# Peer review of "Silencing KRIT1 Partially Reverses the Effects of Disturbed Flow on the Endothelial Cell Transcriptome"

_ijms, 2025, doi:10.3390/ijms26094340_

Round 1

Reviewer 1 Report

Comments and Suggestions for Authors

The manuscript entitled ‚Silencing KRIT1 Partially Reverses the Effects of Disturbed Flow on the Endothelial Cell Transcriptome‘ by Amelia Meecham et al., identified a surprising novel function for Krit1 within the CCM complex in endothelial cells. The authors provide molecular evidence that Krit1 is required for the atheroprone driven phenotype under disturbed flow. Interestingly they found that depletion of krit1 dis-inhibits the klf2/klf4 induction under oscillatory shear stress and restore the Klf2/4 mediated vasoprotective program, effects that are similar to laminar shear stress.

While the results are very interesting and novel, I must emphasize that the manuscript is badly written.

Just a few examples:

  1. ….. In this study, we tested the hypothesis that silencing KRIT1 would promote vasoprotective transcriptional responses in endothelial cells exposed to disturbed flow6. There is no way to add a reference for a hypothesis that authors want to test.
  2. The shortcutting for many words is used many times in the text, e.g PS, OS, siKRIT1.
  3. In addition cells can not be…… ‘subject to static conditions’…..the sentence is incorrect.

Other comments

1.The graphical abstract is confusing. The authors should use the same colors with OS conditions downregulating Klf2/4 thus blocking the protective gene program while blocking Krit results in upregulation of Klf4/2 and therefore inducing the protective gene program.

2.Figure 2B the significance in static and PS conditions between ctr and krit1 siRNA is missing. These should be added.

  1. Do the HUVECs overexpressing Krit1 have similar transcriptomic changes as disturbed FSS? Most probably there are other transcriptomic studies that have done that and the authors could just reefer to published data.

4.Do cells depleted for krit1 under oscillatory FSS behave morphologically as ECs under laminar flow (more elongated or aligned parallel to the direction of flow?). I think this is easy to perform and it will add a complementary approach to the transcriptomic data in the same time proving the  gene sets changes.

Comments on the Quality of English Language

see above

Author Response

Comments regarding writing style:

We acknowledge Reviewer 1's comment regarding the writing quality of the manuscript and recognize the importance of clear and effective communication in scientific writing. We appreciate the reviewer taking the time to point out specific instances where the writing could be improved, which are addressed below.

The reviewer commented regarding the sentence “In this study, we tested the hypothesis that silencing KRIT1 would promote vasoprotective transcriptional responses in endothelial cells exposed to disturbed flow”, with the observation “There is no way to add a reference for a hypothesis that authors want to test". In response to this comment, the sentence in the revised manuscript has been edited and the reference in question has been removed (Page 6).

Reviewer comment: “The shortcutting for many words is used many times in the text, e.g PS, OS, siKRIT1”.

We thank the reviewer for highlighting this aspect of our writing style. To improve readability and clarity, we have implemented the following changes in the revised manuscript. The acronyms PS (pulsatile shear) and OS (oscillatory shear) have been spelled out in the main body of the text at each instance of their use. However, for the sake of brevity and visual clarity, these acronyms have been retained within the figures and defined in each figure legend. Regarding the acronym siKRIT1 (small interfering RNA targeting KRIT1), we have opted to retain this nomenclature of “si” preceding the target gene name is a commonly used abbreviation when referring to siRNA-mediated knockdown of a gene. We have confirmed that all acronyms, including siRNA and siKRIT1, are defined at their first appearance within the manuscript.

Reviewer comment: "In addition cells can not be…… ‘subject to static conditions’…..the sentence is incorrect."

This sentence has been carefully reviewed and corrected in the revised version to read “subjected to static conditions” (Page 7).

The Graphical Abstract:

Reviewer comment: “The graphical abstract is confusing. The authors should use the same colors with OS conditions downregulating Klf2/4 thus blocking the protective gene program while blocking Krit results in upregulation of Klf4/2 and therefore inducing the protective gene program. “

The reviewer found the graphical abstract confusing and suggested using the same colors for OS conditions (both now blue), downregulating Klf2/4 and blocking KRIT1, resulting in upregulation of Klf4/2. In response to the reviewer's suggestion, we have revised the graphical abstract. The color scheme has been modified so that control cells and KRIT1 knockout cells are now represented using the same color. This change visually emphasizes the comparison between control and KRIT1 knockdown under oscillatory shear stress (OS) conditions (Page 3).

Figure 2B (likely Figure 1B):

Reviewer comment: “the significance in static and PS conditions between ctr and krit1 siRNA is missing. These should be added”

The reviewer pointed out that the statistical significance for Static and PS conditions between siCtrl (“ctr”) and siKRIT1 was missing in Figure 2B, suggesting that these should be added. We believe the reviewer is referring to Figure 1B, which presents the mRNA expression levels of KLF2 and KLF4 following KRIT1 knockdown under different flow conditions. In the revised manuscript, we have added the appropriate p-values to Figure 1B for the comparisons between the siSCR and siKRIT1 groups under all conditions (Page 13).

Comparative RNAseq Data

Reviewer Comment: Do the HUVECs overexpressing Krit1 have similar transcriptomic changes as disturbed FSS? Most probably there are other transcriptomic studies that have done that and the authors could just reefer to published data.”

            We appreciate the reviewer's suggestion to investigate whether HUVECs overexpressing KRIT1 exhibit similar transcriptomic changes to those observed under disturbed flow shear stress (FSS). Unfortunately, this relevant experiment has not been conducted or published, and it could represent an important direction for future research.

            Reviewer 1 Comment: “Do cells depleted for krit1 under OS behave morphologically as ECs under laminar flow (more elongated or aligned parallel to the direction of flow?). I think this is easy to perform and it will add a complementary approach to the transcriptomic data at the same time proving the gene sets changes”.

We appreciate the reviewers’ suggestions for additional experiments; however, the present short paper is focused on the genome-wide transcriptomic effects of silencing KRIT1 in disturbed flow. Thus, these interesting questions are beyond the scope of the present work. 

Reviewer 2 Report

Comments and Suggestions for Authors

Authors have made a significant work to investigate the effect of loss of function by KRIT1 and oscillatory shear stress on endothelial transcriptome profile. It is also interesting to look at the transcriptome profile between pulsatile and oscillatory shear stress groups at the basal level (siSCR.os Vs siSCR.PS), since the data is available. The primary focus of the biological pathway analysis would have been on siSCR.OS Vs siKRIT1.OS

Please find the comments below

Major

  1. The information on the role of KRIT1 at endothelial junction and Cerebral cavernous malformations (CCMs) was not enough in the introduction
  2. Why were the results compared between siSCR.PS and siKRIT1 OS? This would only show these two groups' similarities, but the accurate comparison should be between siSCR.OS and siKRIT1.OS, which would provide differential expression between these two groups and the effect of KRIT1 in these biological processes.

 I would like to see the differential expression of these biological pathways (Inflammation, integrin signaling, Cell adhesion, angiogenesis, hypoxia, and shear stress) in siKRIT1.os Vs SiSCR.os. This comparison would make more sense when you are investigating the loss of function due to KRIT1.

  1. What is the expression profile for Inflammatory genes like MCP1, VCAM1, and ICAM1
  2. I would like to see the protein expression of the KLF2 and KLF4 to siKRIT1 along with RTPCR confirmation, which would provide additional confirmation of KRIT1 silencing.
  3. Figure 1D: It would be much better to represent the comparative log fold change values of KLF2, KLF4, and NOS3 between controls (siSCR OS vs siSCR.PS) and siKRIT1 in a tabular form from RNAseq data
  4. The functional role of KRIT1 in endothelial cells has been demonstrated to affect vascular permeability. Did the authors consider looking at the junctional proteins like E-cadherin, ZO-1, JAM-A, and any claudins? Also, HEG1 (anchors KRIT1 on endothelial cells) in the medium in siKRIT1?

Minor

  1. Provide the list of primers and their sequences

Author Response

Improvement to Introduction:

Reviewer comment: “The information on the role of KRIT1 at endothelial junction and Cerebral cavernous malformations (CCMs) was not enough in the introduction”

The reviewer commented that the information on the role of KRIT1 at endothelial junctions and in Cerebral Cavernous Malformations (CCMs) was not sufficient in the introduction. We thank the reviewer for this valuable suggestion and we have added a new paragraph to the introduction that elaborates on the role of KRIT1 and provides more detailed information about Cerebral Cavernous Malformations (CCMs). For the reviewer's convenience, this newly added paragraph has been highlighted in the revised manuscript (Page 6).

Comparing Results siSCR.OS and siKRIT1.OS:

Reviewer Comment: “Why were the results compared between siSCR.PS and siKRIT1 OS? This would only show these two groups' similarities, but the accurate comparison should be between siSCR.OS and siKRIT1.OS”

Our original explanation and annotation in the relevant figure may not have been sufficiently clear. The results presented in the figure are a comparison of the siSCR.OS and siKRIT1.OS to establish the effect of KRIT1 on the transcriptome of HUVEC in disturbed flow. The comparison with the siScrPS and siKRIT1 OS is included to emphasize that the effect of silencing KRIT1 under disturbed flow alters the transcriptome to more closely resemble the vasoprotective state induced by pulsatile flow. We have revised the figure legend to provide a more detailed explanation of the comparisons being made (Page 14).

Expression Profile for Inflammatory Genes:

Reviewer Comment: “What is the expression profile for Inflammatory genes like MCP1, VCAM1, and ICAM1”

We thank the reviewer for their interest in these markers of endothelial cell activation and inflammation. In response, we have added a table to the revised manuscript that presents the expression levels of these and other endothelial cell markers across the different experimental conditions (supplementary table 2). This table provides quantitative data on the impact of KRIT1 silencing and flow pattern on the expression of key inflammatory mediators.

All the raw and processed RNA-seq data, have been made publicly available as part of the submission (Page 8), allowing for further examination and exploration of our findings. The inclusion of this specific data on inflammatory genes directly addresses the reviewer's question and provides additional evidence for the potential vasoprotective effects of silencing KRIT1 under oscillatory sheer stress.

Regarding Protein Expression:

Reviewer Comment: “I would like to see the protein expression of the KLF2 and KLF4 to siKRIT1 along with RTPCR confirmation, which would provide additional confirmation of KRIT1 silencing.”

While we appreciate the reviewer's suggestion for strengthening our study with protein-level validation, we now cite literature that shows that to the existing literature that supports the efficacy of our siKRIT1 approach. Specifically, the paper by Glading et al. (2007) demonstrated that KRIT1 siRNA results in ~70% reduction in KRIT1 protein levels in HUVECs. Our study utilized the same siRNA-mediated approach to silence KRIT1 in HUVECs for a comparable duration (48 hours) with comparable transcript depletion.

Regarding Figure 1D:

Reviewer Comment: “It would be much better to represent the comparative log fold change values of KLF2, KLF4, and NOS3 between controls (siSCR OS vs siSCR.PS) and siKRIT1 in a tabular form from RNAseq data”

The reviewer suggested that it would be much better to represent the comparative log fold change values of KLF2, KLF4, and NOS3 between controls (siSCR OS vs siSCR.PS) and siKRIT1 in a tabular form from the RNA-seq data. In the revised manuscript, we have included these data in Supplementary Table 2.

The Functional Role of KRIT1 in Endothelial Cells:

Reviewer Comment: “The functional role of KRIT1 in endothelial cells has been demonstrated to affect vascular permeability. Did the authors consider looking at the junctional proteins like E cadherin, ZO-1, JAM-A, and any claudins? Also, HEG1 (anchors KRIT1 on endothelial cells) in the medium in siKRIT1?”

We examined the expression of mRNAs encoding junctional proteins like E-cadherin, ZO-1, JAM-A, and various claudins. Our analysis revealed no statistically significant differences in the expression of VE-cadherin, ZO-1, JAM-A, or several claudin mRNAs between the siSCR control group and the siKRIT1 knockdown group under both oscillatory shear stress (OS) and pulsatile shear stress (PS) conditions. This observation is consistent with the findings of Glading et al. JCB 2007 and Lopez-Ramirez et al. JEM 2017, who reported that silencing or inactivating KRIT1 does not impact the mRNA expression levels of junctional proteins. Instead, the alterations are more closely linked to post-translational modifications or the disruption of β-catenin localization at cell-cell junctions.

We appreciate the reviewers’ suggestions regarding the assessment of HEG1 levels in the culture media after silencing KRIT1. However, this short paper focuses specifically on the genome-wide transcriptomic effects of KRIT1 silencing in disturbed flow. As such, these interesting questions fall outside the scope of the current study.

      Primer Sequences

            Reviewer Comment: “Provide the list of primers and their sequences”

We appreciate the reviewer's request for the list of primers and their sequences. This information is provided in Supplemental Table 1. This table includes the gene target, forward, and reverse primer sequences.

Round 2

Reviewer 1 Report

Comments and Suggestions for Authors

The authors have addressed all my concerns, therefore the manuscript should be accepted in the present form.

Nevertheless, I still have one request:

lines 139-141

The sentence is incorrect: Additional control plates of siSCR- and siKRIT1-transfected HUVECs were subjected to static conditions (no flow) and maintained in the same culture medium and conditions described above in an adjacent incubator. 

should be replaced with:

Additional control plates of siSCR- and siKRIT1-transfected HUVECs were grown in static conditions and maintained in the same culture medium as described above.

Reviewer 2 Report

Comments and Suggestions for Authors

The authors have made sufficient changes and explanations to the comments. I have no further comments at this time.